# Galactomannan-Antigen Testing from Non-Directed Bronchial Lavage for Rapid Detection of Invasive Pulmonary Aspergillosis in Critically Ill Patients: A Proof-of-Concept Study

**DOI:** 10.3390/diagnostics13061190

**Published:** 2023-03-21

**Authors:** Kathrin Rothe, Miriam Dibos, Stefanie J. Haschka, Roland M. Schmid, Dirk Busch, Sebastian Rasch, Tobias Lahmer

**Affiliations:** 1Institute for Medical Microbiology, Immunology and Hygiene, School of Medicine, Technical University of Munich, 81675 Munich, Germany; 2Institut Für Laboratoriumsmedizin, Medizinische Mikrobiologie und Technische Hygiene München Klinik, Sektion Mikrobiologie, 81377 Munich, Germany; 3Department of Internal Medicine II, School of Medicine, Technical University of Munich, 81675 Munich, Germany

**Keywords:** invasive pulmonary aspergillosis, galactomannan, BAL-fluid, non-directed bronchial lavage

## Abstract

Invasive pulmonary aspergillosis is associated with high mortality. For diagnosis, galactomannan-antigen in serum and bronchoalveolar lavage fluid is recommended, with higher sensitivity in bronchoalveolar lavage fluid. Because of invasiveness, bronchoalveolar lavage might be withheld due to patients’ or technical limitations, leading to a delay in diagnosis while early diagnosis is crucial for patient outcome. To address this problem, we performed an analysis of patient characteristics of intubated patients with invasive pulmonary aspergillosis with comparison of galactomannan-antigen testing between non-directed bronchial lavage (NBL) and bronchoalveolar lavage fluid. A total of 32 intubated ICU patients with suspected invasive pulmonary aspergillosis could be identified. Mycological cultures were positive in 37.5% for *A. fumigatus*. Galactomannan-antigen in NBL (ODI 4.3 ± 2.4) and bronchoalveolar lavage fluid (ODI 3.6 ± 2.2) showed consistent results (*p*-value 0.697). Galactomannan-antigen testing for detection of invasive pulmonary aspergillosis using deep tracheal secretion showed comparable results to bronchoalveolar lavage fluid. Because of widespread availability in intubated patients, galactomannan-antigen from NBL can be used as a screening parameter in critical risk groups with high pretest probability for invasive aspergillosis to accelerate diagnosis and initiation of treatment. Bronchoalveolar lavage remains the gold standard for diagnosis of invasive aspergillosis to be completed to confirm diagnosis, but results from NBL remove time sensitivity.

## 1. Introduction

Aspergillus is a ubiquitous saprophytic environmental fungus that causes human disease by inhalation or ingestion of airborne conidia, which in healthy individuals are quickly removed by mucociliary clearance and alveolar macrophages [1]. The most common species of Aspergillus causing invasive disease are *Aspergillus* (*A.*) *fumigatus*, *A. flavus*, *A. niger*, *A. terreus* and *A. nidulans*, with *A. fumigatus* accounting for the majority of cases of invasive aspergillosis. Mortality associated with invasive aspergillosis exceeds 50% [2].

Classical risk factors for invasive aspergillosis in patients include hematological malignancy, prolonged neutropenia, immunosuppressive therapy, advanced AIDS or advanced neoplasia, altered lung function such as chronic obstructive pulmonary disease (COPD) and liver failure or liver cirrhosis [3]. Further risk factors are allogeneic stem cell transplantation, intensive care unit (ICU) admittance and influenza/COVID-19. Incidence of invasive aspergillosis is increasing because of rising numbers of patients with immunosuppressive treatment, intensive chemotherapy and stem cell transplantation [4,5]. In particular, patients without classical risk factors for invasive aspergillosis, as defined by the European Organization for Research and Treatment of Cancer/Invasive Fungal Infections Cooperative Group and the National Institute of Allergy and Infectious Diseases Mycoses Study Group (EORTC/MSG), are at risk of delayed diagnosis and high mortality rates [3,6].

The mortality rate of invasive aspergillosis is decreasing due to more effective therapy [7], but early initiation of adequate treatment remains crucial. Thus, establishing the diagnosis of invasive aspergillosis at an early stage improves patient outcomes [8].

For screening purposes and diagnosis of invasive aspergillosis, galactomannan (GM)-antigen tests in serum and bronchial lavage are recommended.

GM is a cell wall component of Aspergillus species which is excreted by the fungus during growth and therefore is correlated with fungal load. Sensitivity of GM-antigen testing is significantly lower in non-neutropenic patients [3] as neutropenia could cause an increased fungal load [9,10]. Sensitivity and specificity are higher in bronchoalveolar lavage (BAL) than in serum [6], which has been attributed to the local infection [11] as endothelial penetration is preceding detectability of GM in serum [12].

Therefore, GM-detection in BAL is a valid test to confirm or rule out invasive pulmonary aspergillosis (IPA) with a sensitivity and specificity of over 90% [3,13].

According to EORTC/MSG-criteria for invasive aspergillosis (categories: proven, probable, possible), a definitive diagnosis can be established by culture of Aspergillus in a specimen obtained from a normally sterile and clinically abnormal site. For probable aspergillosis, recovery of aspergillus in a respiratory specimen as mycological criterion is needed. Here, the acknowledged EORTC/MSG-grading system ranks a positive GM-test as an independent valid mycological criterion, as microbiological cultures can sometimes be false-negative, highlighting the diagnostic value of this test [14].

Frequently, because of the invasiveness of the procedure, gaining BAL fluid is withheld either due to patients’ conditions or due to personnel or technical limitations, which could also be observed during the COVID-19 pandemic. This leads to a significant delay in diagnosis. Therefore, in clinically unstable patients and vital treatment indication, diagnosis has to be established only on clinical and/or radiological grounds. However, antifungal therapy might reduce fungal burden, subsequently leading to false-negative results of GM-antigen testing and mycological culture [15,16], hampering the establishment of the correct diagnosis later.

Widespread availability of diagnostic measures could lead to improved detection and treatment of invasive aspergillosis, resulting in improved patient outcomes. Non-directed bronchial lavage (NBL) is an easy to gain respiratory specimen in intubated patients that are hospitalized, especially in ICUs.

During the COVID-19 pandemic, owing to the risk of aerosolization, only a restricted role for bronchoscopy with BAL was recommended, and the feasibility of NBL as a diagnostic tool was presented in different studies [17,18]. This led to the implementation of the NBL as an adequate alternative to BAL in several national and international guidelines and recommendations [19,20].

We therefore analyzed the diagnostic value of GM-detection in deep NBL, comparing results of GM-antigen-indices in BAL fluid and NBL specimen in intubated patients with probable invasive aspergillosis in a non-COVID-19 cohort.

## 2. Materials and Methods

During the proof-of-concept study period, 32 intubated ICU patients with high clinical risk for invasive aspergillosis were identified. According to local standard procedures for all patients under high suspicion for IPA, a computed tomography (CT) of the lung at ICU admission, serum-GM-testing and GM-testing from BAL fluid were performed. Additionally, in all patients, a NBL using a closed system was used to obtain material. NBL was gained shortly before BAL. BAL/NBL were performed within the first 24 h after ICU admission.

GM detection (PlateliaTM Aspergillus Ag, Bio-Rad Laboratories, Munich, Germany) was performed in BAL-fluid, NBL and in serum samples. Results were reported as optical density index (ODI) with a cut-off of >1.0 for BAL/NBL and >0.5 for serum samples [3].

### Statistics

IBM SPSS Statistics 23 (SPSS, Inc., Chicago, IL, USA) was used for statistical analyses in this study. For descriptive statistics, means and standard deviations were calculated for normally distributed continuous data. To compare other variables, t-tests for normally distributed paired samples were performed, with a *p*-value below 5% (*p* < 0.05) indicating statistical significance.

## 3. Results

During the study period, 32 ICU-patients with high clinical suspicion for IPA were identified. The mean age was 63 ± 15 years and male and female patients were equally represented. Indication for ICU admission in all cases was pneumonia with septic shock (the mean APACHE II score was 23 ± 4, mean SOFA Score 12 ± 3). The mean ICU stay was 22 ± 9 days and the mortality rate was 68%.

All patients had to be intubated at ICU admission. Patient characteristics are presented in Table 1. A total of 23 patients (65.2%) had underlying haemato-oncological diseases such as acute myeloid leukemia (12%), acute lymphoid leukemia (16%), non-Hodgkin lymphoma (19%), multiple myeloma (16%) or myelodysplastic syndrome (9%) and also suffered from severe neutropenia at the time of diagnosis of IPA. Of those 23 patients with underlying haemato-oncological diseases, 15 patients received stem cell transplantation (nine allogenic stem cell transplantations, six autologous stem cell transplantations). In the remaining patients, predisposing pulmonary or hepatic risk factors were present: COPD (9%), liver cirrhosis (12%) and pancreatitis (7%). All patients with COPD had a history of corticosteroid therapy at ICU admission, which also posed a risk for IPA. A total of 17 out of the 23 hematological patients received prophylactic antifungal therapy, with 6 receiving antifungal therapy, at ICU admission. The patients without hematological malignancies did not receive antifungal therapy until the diagnosis of IPA.

All patients had a CT scan of the lung that showed findings consistent with IPA. As far as applicable, for the study population in this special ICU patient collective, risk stratification on the basis of EORTC/MSG-criteria was performed. According to EORTC/MSG-criteria, in the presence of host factors (such as recent history of neutropenia, receipt of allogeneic stem cell transplant and prolonged use of corticosteroids), clinical criteria (including classical findings in the CT scan) and mycological criteria (cytology, direct microscopy, culture or GM-Antigen detection in serum or BAL as an indirect test), probable invasive aspergillosis had to be considered in all cases. Therefore, the selected patients represented a critical risk group with high pretest probability for IPA.

In all patients, BAL was performed to gain lavage fluid explicitly from the suspected lesions previously defined by CT scan. Moreover, NBL was gained by deep tracheal suction from all intubated patients shortly before BAL. In 12 patients (37.5%), mycological cultures from BAL fluid used for GM-antigen-testing were positive yielding *A. fumigatus* in all cases. For all patients, GM-antigen-testing in BAL fluid and NBL using an ODI-value cut-off of >1.0 showed positive results. In the study population, quantitative results of GM-antigen testing for BAL (ODI 3.6 ± 2.2) and NBL (ODI 4.3 ± 2.4), were not statistically different (*p*-value = 0.697) (Table 2).

## 4. Discussion

GM-detection in BAL is a valid test to confirm or rule out IPA with a sensitivity and specificity of approximately 90% using an ODI-cut-off of ≥0.8 [10]. When applying an ODI-value cut-off of 0.5, a sensitivity of 88–90% and a specificity of 81–95% was reported [3,6].

GM-detection represents one gold standard in diagnosis of IPA [3,13] and it is considered a valid microbiological criterion according to EORTC/MSG-definitions. To establish a diagnosis of probable invasive aspergillosis according to these definitions, clinical, microbiological and also host criteria must be present simultaneously [14].

However, not every hospital setting can provide bronchoscopy for every patient at all hours due to technical, personnel or patient-dependent limitations. The invasiveness of the procedure to gain BAL fluid for GM-antigen testing therefore hampers its universal application, resulting in a delay in diagnosis or the necessity to rely on clinical or radiological findings alone, which can be unspecific. Moreover, as timely administration of adequate antifungal therapy is associated with improved patient outcomes [8], every diagnostic criterion should be addressed to improve detectability of IPA. The present study shows that GM-antigen testing in deep NBL specimen and BAL fluid shows similar results with comparable ODI values in a critical risk group with high clinical suspicion and high pretest probability for pronounced invasive IPA. We therefore propose using GM-antigen testing from deep NBL specimens as an emergency screening test in suspected IPA whenever BAL is not instantly available.

That NBL, and even tracheal secretion (TS), materials may be used for detection of IPA could be observed during the COVID-19 pandemic. For COVID-19 associated pulmonary aspergillosis (CAPA), NBL and TS were included in the recent guidelines [20,21].

However, extended diagnostic measure such as BAL remain the gold standard for diagnosis of IPA and should always be completed subsequently, adapted to structural circumstances and each patient’s individual factors. As time sensitivity is an important factor for the diagnosis of IPA, the proposed GM-screening from deep TS can improve patient management by establishing diagnosis at an early stage. Additional diagnostics, which sometimes can only be provided by transferring the patient to another hospital with widespread availability of BAL, can be completed without delay in the initiation of therapy.

This study has several limitations. It is a single-center study based on a proof-of-concept design. Another limitation of the present study is that only patients with high pretest probability for pronounced invasive IPA were clinically selected, resulting in highly positive GM-results (mean ODI > 3). To further elucidate the clinical benefit of this screening test, lower ODI-values also have to be considered to be able to set a reliable cut-off to use in different settings.

GM-antigen testing for different aspergillus species shows variable sensitivity and specificity [22], and piperacillin-tazobactam therapy was shown to cause false-positive results, as it is derived from natural compounds produced by the genus Penicillium [23]. Sensitivity of GM-antigen testing might vary, especially outside the classical risk population, and results from antigen-testing in NBL might be unspecific. Influenza and COVID-19 were recently identified as independent risk factors for IPA in ICU-patients, leading to high mortality even in non-immunocompromised patients [24]. Influenza-associated IPA and CAPA are reported with increasing frequency but might be underdiagnosed as the diagnosis is not primarily suspected in previously healthy patients without neutropenia [24]. This highlights the importance of widely available rapid diagnostic measures for reliable timely detection of IPA. Still more research is needed on GM-testing from NBL, addressing patients without immunosuppression or underlying hematological diseases to gain expertise in these populations in which IPA is reported with increasing frequency.

## 5. Conclusions

GM-antigen testing for detection of IPA using deep NBL showed comparable results to GM-antigen testing in BAL fluid in our cohort. NBL is often universally available in intubated patients, easier to gain than BAL and therefore offers a valuable tool for timely diagnostic workup and establishment of diagnosis in suspected invasive aspergillosis in critical risk groups.

Our study addresses only a critical risk group with high clinical suspicion and high pretest probability for pronounced invasive IPA. In patients without classical risk factors or without defined clinical suspicion for IPA, further studies comparing BAL versus NBL are needed.

## Figures and Tables

**Table 1 diagnostics-13-01190-t001:** Clinical characteristics of the cohort including risk/host factors for invasive aspergillosis.

	All Patients (*n* = 32)
Age (years)	63 ± 15
Male gender *n*; (%)	16; (50)
APACHE II Score	23 ± 4
SOFA Score	12 ± 3
Reason for ICU admission *n*; (%)	
- Septic shock	32; (100)
- Pneumonia	32; (100)
Underlying disease *n*; (%):	
− Hematological malignancy	23; (65.2%)
- Acute myeloid leukemia	4; (12)
- Acute lymphoid leukemia	5; (16)
- Non-Hodgkin lymphoma	6; (19)
- Multiple myeloma	5; (16)
- Myelodysplastic syndrome	3; (9)
- Stem cell transplant (SCT) recipients (*n*):	
- allogenic SCT	9
- autologous SCT	6
− Other	
- COPD/Steroid therapy	3; (9)
- Liver cirrhosis	4; (12)
- Pancreatitis	2; (7)
Severe neutropenia *n*; (%):	23; (72)
Serum galactomannan (ODI; mean)	1.9 ± 0.6
− CT scan findings (*n* = patients)	
− Pulmonary infiltrates	32
− Consolidations	21
− Nodules	12
− Masses	11
− Hemmorrhagic infarction	6
− Cavities/Air-meniscus sign	3
ICU stay (days)	22 ± 9
Mortality rate *n*; (%)	22; (68)

Abbreviations: COPD, chronic obstructive pulmonary disease; ICU, intensive care unit; APACHE II, Acute Physiology and Chronic Health Evaluation; SOFA, Sequential Organ Failure Assessment Score. Data are presented as mean ± standard deviation.

**Table 2 diagnostics-13-01190-t002:** Results of galactomannan-antigen testing from bronchoalveolar lavage fluid and tracheal secretion.

	Galactomannan from BAL:	Galactomannan from NBL:	*p*-Value
Mean galactomannan (ODI)	3.6 ± 2.2	4.3 ± 2.4	0.697
Probable invasive aspergillosis (*n*)	32		
Positive galactomannan result (*n*)	32		
Cultural microbiological evidence (*n*)	12 (all *A. fumigatus*)		

Abbreviations: BAL, bronchoalveolar lavage; NBL, non-directed broncho lavage; ODI, optical density index. Data are presented as mean ± standard deviation. A t-test for normally distributed paired samples was performed, with *p* < 0.05 indicating statistical significance.

## Data Availability

The data presented in this study are available on request from the corresponding author. The data are not publicly available due to privacy restrictions.

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
