# Peer review of "Galactomannan-Antigen Testing from Non-Directed Bronchial Lavage for Rapid Detection of Invasive Pulmonary Aspergillosis in Critically Ill Patients: A Proof-of-Concept Study"

_diagnostics, 2023, doi:10.3390/diagnostics13061190_

Round 1

Reviewer 1 Report

This study presented a comparison of galactomannan-antigen testing between non-directed bronchial lavage (NBL) and bronchoalveolar lavage fluid in intubated ICU patients with suspected invasive pulmonary aspergillosis. They concluded deep NBL showed comparable results to GM-antigen testing in BAL fluid. However, the results are incomplete and inconclusive. My major concerns are as follows:

Comments:

1. How about the GM testing in serum from these patients? Since all these included patients had a high probability of IPA, it is likely the GM testing in serum was also positive, which seems a more convenient and less risky means to identify probable IPA.

2. Please clarify the criteria of “high clinical suspicion for pronounced invasive aspergillosis” in Line 94.

3. Were these patients included in this study consecutively or not? The timing of taking specimens should also be mentioned, before or after any use of antifungal medications.

4. CT findings of these patients should be described, pulmonary infiltrates? The number of nodules? Any signs (halo, cavity)?

5. Please check the spelling mistakes, such as “patent” in Line 88.

Author Response

Dear editors, dear reviewers,  

first of all we would like to thank you for the opportunity to revise our manuscript to a quality suitable for publication in Diagnostics. As requested, we respond to each issue raised by the reviewer in turn. A revised version with highlighted track changes in red is also submitted.

Thank you again considering our manuscript for publication in Diagnostics.

We appreciate your time and look forward to your response.

Kind regards,

Tobias Lahmer, M.D.

Dear reviewer 1,

Thank you for your helpful comments to improve the quality of our manuscript. In the following we respond to your comments.

Reviewer 1:

This study presented a comparison of galactomannan-antigen testing between non-directed bronchial lavage (NBL) and bronchoalveolar lavage fluid in intubated ICU patients with suspected invasive pulmonary aspergillosis. They concluded deep NBL showed comparable results to GM-antigen testing in BAL fluid. However, the results are incomplete and inconclusive. My major concerns are as follows:

Comments:

  1. How about the GM testing in serum from these patients? Since all these included patients had a high probability of IPA, it is likely the GM testing in serum was also positive, which seems a more convenient and less risky means to identify probable IPA.

Thank you for this helpful comment. We stated already GM testing in the manuscript, however, we have not included the values, which are now included in table 1. Although, serum testing is less invasive, getting material from the bronchoalveolar system is necessary for many reasons. Species identification and in some cases proof of antifungal resistance can be crucial. In patients without hematological malignancies and neutropenia, serum testing is in most cases negative and BAL or NBL are the only possibilities to get diagnosis of IPA. In ICU patients BAL/NBL are standrard procedures with low risk of complications. Furthermore, immunocompromised patients often need BAL/NBL not only to find IPA but also to diagnose bacterial or viral Co- or Superinfections.  

  1. Please clarify the criteria of “high clinical suspicion for pronounced invasive aspergillosis” in Line 94.

We rephrased this section.

  1. Were these patients included in this study consecutively or not? The timing of taking specimens should also be mentioned, before or after any use of antifungal medications.

Thjs is a good point. BAL/NBL was performed within in the first 24hours after ICU admission in all patients. We included this in the methods section. The patients without hematological malignancies have not received fungal therapy or prophylaxis before diagnosis of IPA.

In the patients with hematological malignancies 17 patients received prophylaxsis and 6 antifungal therapy at ICU admission. We stated this in the results section. 

  1. CT findings of these patients should be described, pulmonary infiltrates? The number of nodules? Any signs (halo, cavity)?

Thanks for your advise, we included the CT findings in table 1.  

  1. Please check the spelling mistakes, such as “patent” in Line 88.

We revised the manuscript accurately

Once again we would like to thank you for the possibility to revise our manuscript. We hope you are satisfied with the changes we made and looking forward for publication in your journal.

Kind regards

Tobias Lahmer

Reviewer 2 Report

Rothe K. and colleagues compared the performance of GM testing in BAL (gold standard) and NBL samples in patients with probable IPA admitted to ICU. Authors report that quantitative GM values in BAL and NBL were not significantly different.

The major limitations of this study, as clearly stated by the Authors, arethe study population (high prevalence of the disease, high PPV of the test) and absence of negative controls, that do not allow to draw conclusions about GM performance in NBL.

I have some comments:

-  in line 102 you report a cut-off of 1.0 for GM, while in line 144 the reported cut-off is 0.5; can you explain?

- you report that CT-scan of the lung showed findings consistent with IPA; can you describe this findings? CT scan was performed before or after ICU admission?

-the introduction should be improved by discussing how NBL was used in previous studies, especially during the COVID-19 pandemic (for example 10.1093/cid/ciaa1298, 10.1093/cid/ciaa1065, 10.1164/rccm.202005-2018LE).

I would shorten the conclusion seciton.

Thank you for your work.

Author Response

Dear editors, dear reviewers,  

first of all we would like to thank you for the opportunity to revise our manuscript to a quality suitable for publication in Diagnostics. As requested, we respond to each issue raised by the reviewer in turn. A revised version with highlighted track changes in red is also submitted.

Thank you again considering our manuscript for publication in Diagnostics.

We appreciate your time and look forward to your response.

Kind regards,

Tobias Lahmer, M.D.

Dear reviewer 2,

Thank you for your helpful comments to improve the quality of our manuscript. In the following we respond to your comments.

Reviewer 2:

Rothe K. and colleagues compared the performance of GM testing in BAL (gold standard) and NBL samples in patients with probable IPA admitted to ICU. Authors report that quantitative GM values in BAL and NBL were not significantly different.

The major limitations of this study, as clearly stated by the Authors, arethe study population (high prevalence of the disease, high PPV of the test) and absence of negative controls, that do not allow to draw conclusions about GM performance in NBL.

I have some comments:

-  in line 102 you report a cut-off of 1.0 for GM, while in line 144 the reported cut-off is 0.5; can you explain?

Thanks for your comment. This was a fault, we rephrased this section.

- you report that CT-scan of the lung showed findings consistent with IPA; can you describe this findings? CT scan was performed before or after ICU admission?

Thanks for your advise, we included the CT findings in table 1. 

CT scan was performed at ICU admission. We included this in the material and methods section.

-the introduction should be improved by discussing how NBL was used in previous studies, especially during the COVID-19 pandemic (for example 10.1093/cid/ciaa1298, 10.1093/cid/ciaa1065, 10.1164/rccm.202005-2018LE).

Referring to your suggestions we implemented a additional section in the introduction

I would shorten the conclusion seciton.

We rephrased this paragrapgh.

Thank you for your work.

Once again we would like to thank you for the possibility to revise our manuscript. We hope you are satisfied with the changes we made and looking forward for publication in your journal.

Kind regards

Tobias Lahmer

Round 2

Reviewer 1 Report

Accept in present form